# Minor Intron Splicing from Basic Science to Disease

**DOI:** 10.3390/ijms22116062

**Published:** 2021-06-04

**Authors:** Ettaib El Marabti, Joel Malek, Ihab Younis

**Affiliations:** 1Weill Cornell Medicine-Qatar, Education City, Doha P.O. Box 24144, Qatar; ete2001@qatar-med.cornell.edu (E.E.M.); jom2042@qatar-med.cornell.edu (J.M.); 2Biological Sciences Program, Carnegie Mellon University in Qatar, Doha P.O. Box 24866, Qatar

**Keywords:** minor introns, U2 introns, U12 introns, minor spliceosome, RNA splicing, disease

## Abstract

Pre-mRNA splicing is an essential step in gene expression and is catalyzed by two machineries in eukaryotes: the major (U2 type) and minor (U12 type) spliceosomes. While the majority of introns in humans are U2 type, less than 0.4% are U12 type, also known as minor introns (mi-INTs), and require a specialized spliceosome composed of U11, U12, U4atac, U5, and U6atac snRNPs. The high evolutionary conservation and apparent splicing inefficiency of U12 introns have set them apart from their major counterparts and led to speculations on the purpose for their existence. However, recent studies challenged the simple concept of mi-INTs splicing inefficiency due to low abundance of their spliceosome and confirmed their regulatory role in alternative splicing, significantly impacting the expression of their host genes. Additionally, a growing list of minor spliceosome-associated diseases with tissue-specific pathologies affirmed the importance of minor splicing as a key regulatory pathway, which when deregulated could lead to tissue-specific pathologies due to specific alterations in the expression of some minor-intron-containing genes. Consequently, uncovering how mi-INTs splicing is regulated in a tissue-specific manner would allow for better understanding of disease pathogenesis and pave the way for novel therapies, which we highlight in this review.

## 1. RNA and Splicing

The view that RNA is simply an intermediate between DNA (the keeper of genetic information) and proteins (the executioners of all cellular functions) has long been challenged by the discoveries that gave RNA its own catalytic functions such as self-splicing introns [1] and the ribonuclease P catalyst [2]. These discoveries provided further evidence for the RNA world theory, which stipulates that life on earth started with RNA. The primary living substance on primitive earth is therefore thought to be RNA or a chemically similar structure, a process that may have started 4 billion years ago. The evidence for this 50 year old theory has grown over the years, and the importance of RNA has grown with it, giving rise to new fields in RNA biology that have expanded toward medical/clinical applications, including experiments done in the early 1990s showing protein production from an mRNA injected into mice skeletal muscle [3] and antibody generation from influenza virus mRNA injection [4], to most recently using mRNA as a vaccine against SARS-CoV-2 [5,6]. Thus, RNA-based therapies are quickly progressing to treat many ailments by targeting various pathways in RNA metabolism, including pre-mRNA splicing, which has garnered increasing attention after a handful of successful applications.

Pre-mRNA splicing was initially discovered in adenoviruses but was later described to occur in all eukaryotes [7]. More than 95% of genes in humans are transcribed as pre-mRNAs containing introns (intervening regions) that interfere with the required sequence continuity of exons (expressed regions) [7,8]. Splicing occurs co-transcriptionally to ensure the removal of introns and subsequent joining of exons, making it an essential and required step for the proper formation of a mature mRNA transcript that could be used as a template for protein synthesis [7,8]. Of note here is that a large number of intron-containing genes are transcribed and spliced yet do not get translated into a protein but rather have key functions as noncoding RNAs. Broadly, the sequences involved in splicing include a 5′ splice site (5′ ss), a branch point sequence (BPS), and a 3′ splice site (3′ ss), and splicing is carried out as a two-step reaction, starting with 5′ splice site cleavage, generating a lariat, and ending with 3′ splice site cleavage and exon ligation [8]. This reaction is catalyzed by the spliceosome, an RNA-protein complex that requires ATP and Uridyl-rich small ribonucleoproteins (U snRNPs) [9].

## 2. The Other Side of the Splicing Coin—An Overview

The majority of intron splicing relies on the ubiquitous major spliceosome formed by the U1, U2, U4, U5, and U6 snRNPs [10,11]. These introns are loosely termed U2 introns and are referred to as such hereafter. Interestingly, eukaryotes carry in their genomes a less common type of introns termed minor introns (mi-INTs) or U12 introns (in this review, we use both terms interchangeably). The number of mi-INTs vary between species with humans having one of the largest number (~750) of minor-intron-containing genes (miGs) [12,13,14,15]. The mere existence of mi-INTs is remarkable for many reasons. Rather than using the highly abundant and ubiquitous major spliceosome, mi-INTs utilize a specialized spliceosome composed of a distinct set of snRNPs (U11, U12, U4atac, U5, and U6atac) [12,13,14,15]. The first mi-INTs were identified due to the unusual dinucleotide termini (AU at the 5′ end and AC at the 3′ end of the intron, compared to the conventional GU and AG, respectively) [12,16]. With the discovery of more mi-INTs, it became clear that the AU-AC dinucleotide at the intron boundaries is not what distinguishes U12 from U2 introns as most mi-INTs were shown to have a GU-AG dinucleotide but possess other unique features that set them apart from U2 introns [17,18]. These features played an important role in computationally identifying mi-INTs and include sequences within the introns as well as the conservation of their 5′ ss and BPS over 100 million years [19]. The peculiarity of mi-INTs however extends beyond their sequence features. In spite of their low abundance, the position of mi-INTs in their host genes is also highly conserved between humans, animals, and plants [14,20,21]. On the other hand, miGs tend to show commonality in their functions and have been coined by Burge et al. as “information processing genes” [19]. These functions include DNA transcription, replication and repair, RNA processing and translation, cytoskeletal organization, vesicular transport, voltage gated ion channel activity, and Ras-Raf signaling [22,23]. These very important functions seemed for a long time to be contradictory to in vitro and in vivo experiments showing slow splicing rates and hence inefficiency of the splicing of mi-INTs, making them bottlenecks for the expression of their host genes [24]. Moreover, data showing that mi-INTs can be experimentally converted to U2 introns with relative ease, as this required only a few mutations in their 5′ ss [17,19,25], adding serious doubt regarding the purpose of minor intron splicing altogether. Lastly, another puzzling feature of mi-INTs is the low abundance of their minor spliceosome (~100-fold compared to the major spliceosome), especially the catalytic component U6atac snRNP [26]. The dilemma of these seemingly contradictory features has been partially resolved as discussed later in this review.

Pinpointing the exact origin of introns in general has been elusive. However, their approximate age has been speculated based on their absence from prokaryotic genomes and presence in most eukaryotic and protist genomes [19,27,28]. Prior to the discovery of the U12 splicing pathway, attempts to explain the origin of introns followed two theories: the “exon theory of genes” and the “protosplice theory”. The former stipulated that introns came into existence to allow exon assortment, giving rise to diverse genes through intron recombination, while the latter suggested that introns invaded eukaryotic genomes and were inserted at protosplice sites [29,30,31,32]. The discovery of U12 introns has challenged both theories as the presence of two distinct types of introns had not been accounted for by either theory. mi-INTs have a nonrandom distribution (clustered in information processing genes), and they occur less frequently in the genome. However, despite our lack of understanding of the origin of U12 introns, it was clear that these introns have appeared early in eukaryotic evolution and have been conserved in many organisms, spanning several kingdoms [20,27,33]. These include plants (*Arabidopsis thaliana*), vertebrates (fish, amphibians, birds, and mammals), insects (*Drosophila melanogaster* and silkworm *Bombyx mori*), cnidarians (jellyfish), protists (*Acanthamoeba castellanii*), and fungi (*Rhizopus oryzae*). Nevertheless, it appears that mi-INTs have also been lost throughout evolution as they are absent in some fungi (*Saccharomyces cerevisiae* and *Schizosaccharomyces pombe*), some protists, and nematode (*Caenorhabditis elegans*) [19,27]. Some evidence suggests that some U12 introns loss has occurred through conversion to U2 introns [17,19]. Again, in light of U12 introns’ apparent disadvantages (slow kinetics, low abundance of minor spliceosome, etc.) and their ability to potentially easily convert to U2 introns, coupled with the extensive evidence of their conservation, and the essential functions of miGs, the conundrum of why mi-INTs even exist ensued. The working model for the why mi-INTs originated and are extremely conserved is that they play an important biological function within the genes that host them, driving their maintenance within miGs and conservation across organisms.

Several lines of evidence support this model of inefficiency for regulatory purposes and challenged the idea that mi-INTs retention is due to an inherent inefficiency in this splicing pathway. It was shown that the splicing inefficiency of mi-INTs resulting in partial or complete intron retention leads by design to specific outcomes, including degradation of their host transcripts by non-sense-mediated decay (NMD) or the exosome, alternative splicing of the transcript, or translation into a truncated protein. These outcomes are in support of mi-INTs retention being a post-transcriptional mechanism regulating miGs expression. Indeed, transcripts retaining mi-INTs were shown to constitute a pool of pre-mRNAs that is on standby to be expressed when required based on cellular needs. This is determined by the level of U6atac snRNP which is regulated through a MAPK signaling pathway. The effectiveness of this regulatory role of mi-INTs lies in the ability of a single intron to regulate the expression of a whole pre-mRNA. Therefore, the overarching function of mi-INTs in hundreds of miGs is suggested to be an expression “molecular switch” [34]. While the molecular switches model implies an all or none regulation, other data suggested a role for mi-INTs in alternative splicing as shown for the splicing factor Srsf10 [35], which is a member of the SR protein family that regulates alternative splicing. Srsf10 autoregulates its splicing through two competing minor and major 5′ splice sites, resulting in a protein-coding or a noncoding transcript, respectively. This autoregulation leads to a dynamic mechanism where the minor spliceosome activity determines *Srsf10* gene expression, and this in turns affects the levels of many other SR proteins. This work highlights a role for the minor spliceosome in regulating alternative splicing of many U2 introns indirectly through regulating the levels of SR proteins in cells [35].

The clinical relevance of splicing is well established, as 90% of disease-causing SNPs lie outside the protein-coding regions and many could have underlying mechanisms that would lead to aberrant splicing [36,37]. Indeed, some models estimate that 62% of disease mutations act by affecting splicing [38]. While it is still unclear what exact proportion of these mutations affects mi-INTs splicing as the targets of these mutations are being uncovered, several known mutations do include components of the minor splicing machinery (snRNPs), mi-INTs splice sites, and loci important for biogenesis and assembly of the minor spliceosome. Such aberrations lead to diseases ranging from neurodevelopmental and neurodegenerative to autoimmune disorders and cancer [39]. While the latter is well established in relation to mis-splicing of U2 introns and less so to that of mi-INTs [37,39], the similarity between the two splicing pathways, in addition to some evidence about alterations of mi-INTs splicing in cancer [40,41,42], provides further support that both U2 and U12 intron splicing may be affected in a similar fashion in cancer.

Mutations within components of the minor intron splicing machinery have been linked to the majority of diseases caused by minor intron splicing alterations. These mutations are predicted to affect hundreds of target genes containing mi-INTs, with the possibility of causing systemic defects in many body tissues. However, despite the ubiquitous nature of these mutations, most of the resulting diseases are tissue specific, indicating the possibility that only a subset of target genes has the potential of causing disease. Fortunately, recent advances in genomics coupled with efforts to understand mi-INTs biology have shed some light on the mechanisms underlying disease tissue specificity and ways to identify target genes. In this review, we discuss this tissue specificity based on our current understanding of mi-INTs splicing and suggest experimental approaches using animal models coupled with CRISPR screens to identify target genes. We also bring to light new advances in the field of mi-INTs through discussing mi-INTs characteristic, the current debate of inefficiency vs. regulation, the fate of retained mi-INTs, and the crosstalk between the minor and major spliceosomes.

## 3. mi-INTs Sequence Characteristics

Initially described as introns with AU-AC termini (Figure 1A), mi-INTs containing these termini were later found to be only a subtype of all U12 introns [12], and the majority of mi-INTs have GU-AG termini [17,18]. Furthermore, mi-INTs could also have other, albeit much rarer, termini including AU-AG, GU-AU, CU-AC, GG-AG, and GA-AG [17,43]. Consequently, mi-INTs classification had to rely on more specific characteristics to distinguish them from U2 introns that have the canonical GU-AG termini. mi-INTs key sequence features lie in the conservation of nucleotide segments in their 5′ ss and BPS immediately upstream of their 3′ ss [12]. The 5′ ss conserved segment includes 8–9 nucleotides (nts) (A/G)UAUCCUUU, and the BPS includes the conserved sequence UCCUUAAC with fewer variations compared to U2 introns [12,19,44]. The splicing of mi-INTs depends on a well-situated 3′ ss creating a functional constraint on the distance between the BPS and the 3′ ss [44]. Thus, this distance is relatively shorter (10–20 nts) in mi-INTs, with the optimal distance being 11–13 nts [44]. Furthermore, mi-INTs lack a polypyrimidine tract which is commonly found in U2 introns (Figure 1A). The splice sites sequences and their conservation in multiple species can be found at https://genome.crg.es/cgi-bin/u12db/u12db.cgi (accessed on 19 May 2021).

mi-INTs sequence features are critical for their recognition, the formation/assembly of the splicing machinery, and adequate splicing through the 5′ and 3′ ss (Figure 1B). This splicing is carried out by the less abundant minor spliceosome, with unique snRNPs and protein composition, and slower splicing kinetics (refer to the spliceosome section for more detail).

## 4. Origin and Evolution: The U12 Splicing Pathway Is as Old as the U2 Pathway

mi-INTs were initially exclusively described in animals [12,16] but were later identified in plants with identical sequence characteristics, dating the origin of the minor spliceosomal pathway to at least one billion years [45], as it is thought to have appeared prior to the divergence of the animal and plant kingdoms. This is supported by the discovery of mi-INTs and the homologous component of the minor splicing pathway (U11, U12 snRNA, and minor spliceosome-specific proteins) in protists and fungi [27]. The presence of components of a minor spliceosomal pathway in protists is evidence that the minor spliceosome is possibly as old as its counterpart, the major spliceosome. The evolution of these splicing pathways could have followed three models: the group II introns parasitic invasion model [19,46,47], the codivergence model [48], or the fission/fusion model [19]. The parasitic invasion model assumes that following endosymbiont invasion of an archaeal organism, descendants of two distinct group II introns invaded nuclear genes sequentially or simultaneously [19,28,46,47]. This may have occurred through repeated lysis of the invading organism (the developing mitochondria) causing seeding of these introns [28,47,49]. The fragmentation of these introns would have led to novel U snRNAs that could utilize proteins of the pre-existing spliceosome (supposedly U2-dependent spliceosome from a prior invasion) to splice these introns [19,47,50,51]. A protosplice site comparison of ancient introns to U2 and U12 type introns in humans and *Arabidopsis thaliana* suggests that primordial spliceosomal introns were of the U2 type [51]. Consequently, based on the parasitic invasion model, it is thought that U2 introns were the first to populate genes, followed by U12 introns [47,51].

However, this model does not explain the existence of “twintrons” in some pre-mRNAs. A twintron is basically a major intron within a minor intron, such as that in the *Drosophila prospero* gene, which encodes a protein that plays an important role in axonal outgrowth and cell specification during the development of the nervous system [52]. Like other twintrons, that in the *prospero* gene provides a unique mechanism of alternative splicing. Remarkably, the switching between U2-dependent and U12-dependent spliceosomes is highly regulated during development and has significant consequences on the function of the gene product. More specifically, U12-dependent splicing of the twintron produces a shorter mRNA that goes on to be translated into a transcription factor with a specific homeodomain for proper DNA binding, whereas the mutually exclusive U2-dependent splicing results in a longer mRNA, that when translated, encodes a different homeodomain with potentially altered DNA-binding specificities [52]. Assuming that spliceosomal introns invaded nuclear genomes, twintrons suggest that U12 introns may not have arisen after U2 introns [48].

Furthermore, while Basu et al.’s findings were based on evidence that U12 introns positions are more conserved than those of U2 introns in humans and *Arabidopsis thaliana* [51], Moyer et al.’s data show that conservation in both pathways is similar, as 9% of U12 introns and 22% of U2 introns have been shown to have conserved position between humans and *Arabidopsis thaliana* [25]. These data argue against a parasitic invasion model in which U2 intron invasion preceded U12 invasion. On the other hand, the codivergence model assumes that following snRNA duplication in a primordial organism, introns, snRNA, and spliceosomal protein components diverged into two pathways [19,48]. Lastly, the fission/fusion model stipulates that following speciation of two different lineages from a primordial organism, the two splicing pathways diverged with each lineage containing one of the systems [19]. These organisms eventually merged, allowing the mixing of their genetic material [19]. Overall, current models support the idea that U12 and U2 introns and their splicing systems emerged around the same time.

Phylogenic reconstructions of the minor and major spliceosome’s snRNAs and proteins show that both spliceosomes existed in the last eukaryotic common ancestor (LECA) [28]. The phylogenic distribution of U12 introns includes plants, vertebrates, insects, Cnidarians, protists, and fungi [19,27]. Nonetheless, they are absent in some fungi (*Saccharomyces cerevisiae* and *Schizosaccharomyces pombe*), some protists, and nematode (*Caenorhabditis elegans*) [19]. The distribution of minor spliceosomal components shows the existence of a minor spliceosomal pathway in certain organisms that are within or below eukaryotic groups, suggesting loss of the minor splicing system during evolution [27]. One example is in fungi (Ascomycetes and Basidiomycetes) which lack minor spliceosomal components as opposed to *Rhizopus* (Zygomycota) which contains proteins components of the minor spliceosome and U12 introns. Consequently, evolutional diversification within eukaryotes may have led to an eradication of U12 introns and the minor spliceosomal system in some organisms [27]. This loss is explained by the “class conversion hypothesis”, which is believed to have occurred through the unidirectional U12 to U2 conversion. This conversion seems to preferentially occur in the GU-AG subtype of U12 introns and is unidirectional (U12 to U2) due to the constraints in the characteristics of U12 intron splice sites sequences [20,25,27]. However, it is important to note that U12 intron loss, albeit slow in vertebrate, seems to be more common than conversion [43]. A comparison of miGs across different organisms suggests that some U12 introns have been replaced by U2 introns, potentially through splice site mutations. For example, while in humans, intron 1 of *PTEN* is a U12 intron; in *C. elegans*, intron 1 is a U2 intron. Recently, a phase 0 surplus in U2 introns and an underrepresentation of phase 0 in U12 introns [25] suggested that class conversion has occurred selectively in phase 0 U12 introns with GU-AG termini, supporting the idea that class conversion (U12 to U2) occurs through subtype switching (AU-AC → GU-AG) [19]. This conversion and loss contributed to the phase biases and low abundance of U12 introns in modern eukaryotic genomes as opposed to their ancestral counterpart.

The evolutionary loss of U12 introns poses a challenge to their intrinsic feature of conservation (phylogenetically and within gene families). This intriguing discrepancy has been partially resolved with new studies suggesting that mi-INTs function is the main driver of conservation in this class of introns.

## 5. The Functional Relevance of Minor Introns

Minor-intron-containing genes have been described as “information processing” as opposed to “operational genes” involved in metabolism, for example [19,22,23]. mi-INTs are enriched in gene families that are involved in DNA replication/repair, transcription, translation, and RNA processing. Other enriched functional groups for miGs include voltage-gated Na and Ca channels, vesicular transport, and cytoskeletal organization [21]. Interestingly, many mi-INTs have been shown to have slow splicing kinetics (~2 fold slower than U2 introns), both in vivo [24,53] and in vitro [13,54], leading to increased mi-INT retention, while all other U2 introns in the same transcript are spliced out efficiently [24]. As the majority of miGs contain a single minor intron per gene (only a small fraction contain two or more) [55], this one intron could be rate-limiting for the expression of the miG. Thus, the nonrandom genomic distribution of mi-INTs, the fact that the majority of them exist as a single intron within miGs, and their inefficient splicing/retention led to speculations that mi-INTs play important regulatory functions. Consequently, mi-INTs were dubbed “post-transcriptional bottlenecks” which would serve to prevent gene overexpression that may be harmful to cells [24]. This is supported by the molecular switches theory [34] as shown in Figure 2A, which stipulates that mi-INTs retention forms a pool of pre-mRNA transcripts that can be rapidly expressed when needed by the cell [34]. Therefore, rather than being breaks, halting gene expression to prevent toxicity, they are considered molecular switches that can be quickly turned on or off to allow or prevent gene expression [34]. This idea was built on evidence of the high instability of U6atac snRNP, a crucial component of the minor spliceosome catalytic core [34]. This instability causes low levels of U6atac in cells leading to low levels of properly spliced mRNA for hundreds of miGs. However, once stabilized, higher U6atac rapidly enhances mi-INTs splicing and miGs expression [34]. One example of this “on-demand” miG expression is shown when U6atac stabilizes by the stress-activated protein kinase, p38MAPK, which activates these molecular switches to allow the expression of genes required to deal with cellular stress [34]. This provides an efficient mechanism through which one intron could regulate the expression of an entire transcript, a rational idea that may explain the conservation of the minor intron splicing system.

More recent work has revealed another facet of mi-INTs as alternative splicing regulators, through the SR protein Srsf10, which itself functions in regulating alternative splicing [35]. One notable feature of the protein Srsf10 is that it autoregulates its expression through a cis regulatory element in its pre-mRNA [35] (Figure 2B). This autoregulation determines the use of competing major and minor splice sites leading to nonproductive or productive transcripts, respectively [35]. However, minor spliceosomal activity seems to partially overrule this autoregulation, as lower minor spliceosomal levels significantly lower the expression of productive Srsf10 transcripts [35], suggesting that minor spliceosomal abundance controls *Srsf10* gene expression by regulating its alternative splicing [35] (Figure 2B). Furthermore, in vivo evidence suggested that the expression of many other SR proteins, which themselves do not contain mi-INTs, correlates with minor spliceosomal levels in 25 mice tissues. CRISPR/Cas9 deletion of the nonproductive Srsf10 transcript led to increased expression of many other SRSF proteins [35], establishing that minor spliceosome abundance/activity controls *Srsf10* expression which in turn changes the expression of other SR proteins [35]. Taken together, these data suggest that the purpose of the minor splicing pathway is to regulate gene expression at the alternative splicing step of both miGs as well as U2 intron albeit indirectly [24,34,35].

## 6. mi-INTs Splicing Is Regulated

Minor-intron-containing genes show differential expression across tissues. Olthof et al. showed that both the number of miGs and their expression levels deferred across 11 mice tissues and 8 human tissues, with the heart and liver, for example expressing the least number of miGs at the lowest levels [56]. Importantly, miGs expression patterns across mice and human tissues were generally conserved [56]. Furthermore, miGs expression signatures showed functional enrichment in a tissue-specific manner [56]. For example, the cerebrum showed higher expression of miGs enriched in voltage-gated ion channel activity [56]. Intriguingly, while earlier data on a few mi-INTs have suggested an overall inefficient minor splicing pathway, recent data from 582 mi-INTs in 11 mice tissues point toward a dynamic regulatory mechanism in tissues as the inefficiency may not apply to all miGs in all tissues [56]. Therefore, mi-INTs splicing seems to be regulated in a tissue-specific manner to allow differential miGs expression. This would allow enrichment in functions that are important for specific tissues [56].

While mi-INTs retention could be an opportunity to generate a pool of unspliced transcripts standing by to be induced, it could also lead to one of the following predictable consequences (Figure 2A): (1) degradation of the host transcript by NMD or the exosome; (2) translation into a truncated protein if the transcript manages to be exported to the cytoplasm and is accessible to the ribosome, as most mi-INTs are expected to have in frame stop codons; and (3) inducing alternative splicing (AS) using cryptic or alternate splice donor, splice acceptor, or both. While the latter consequence is the least studied [21], early studies on twintrons that could switch between minor and major splicing pathways in *Drosophila* [52] and were later confirmed to occur in some human miGs [34], support the notion that mi-INTs provide a platform for AS. Recent work by Olthof et al. proposed that AS of mi-INTs occur more frequently than previously thought, and importantly it occurs in a tissue-specific manner [56]. Taken together, the idea that mi-INTs retention could be considered as a form of alternative splicing that is regulated agrees with the molecular switches mechanism previously described. Finally, the discovery of a high frequency of AS across this class of introns provides new avenues for the discovery of new RNA transcripts, protein isoforms/antigens that may be elevated in disease. These could therefore serve as biomarkers or targets for therapy.

## 7. The Minor Spliceosome: Composition, Assembly, and Evolution

The spliceosome is a multimegadalton ribonucleoprotein (RNP) complex that catalyzes the removal of introns in eukaryotes through two consecutive transesterification reactions [11,14]. The assembly of the spliceosome, which requires the recognition of intron consensus sequences at the 3′ and 5′ ends [11,14], is an organized process that involves a multitude of interactions between five U-rich small nuclear ribonucleoproteins (U snRNPs) and many associated proteins [11,14]. Upon assembly, RNA–RNA, RNA–protein, and protein–protein interactions allow for the catalysis of the splicing reaction; releasing the intron lariat and ligating the exons [11,14].

As described above, most eukaryotes contain two spliceosomes: the major (U2-dependent) spliceosome (reviewed in [11,57]), and the minor (U12-dependent) spliceosome which splices out mi-INTs. The minor spliceosome is made up of the minor snRNPs: U11, U12, U4atac, and U6atac, which are 100-fold less abundant [26,58] than their major spliceosome homologues U1, U2, U4, and U6, whereas U5 snRNP is shared by both spliceosomes. While the snRNPs composition in the two spliceosomes is homologous, there seems to be little sequence conservation between the minor and major snRNAs [50,58]. U11 and U12 sequences are unrelated to those of U1 and U2, respectively [26], while U4atac, U6atac have 40% sequence similarity to human U4 and U6, respectively [58]. Most of these conserved sequences are in regions important for RNA–RNA interaction within the spliceosome (between U6atac and U4atac or U12) or with the pre-mRNA [58]. However, some of them share analogous secondary structures, such as U4 and U6, and U4atac and U6atac interact through a base pairing interaction forming a di-snRNP secondary structure [58]. Furthermore, the U4atac/U6atac di-snRNP interacts with U5 to form a tri-snRNP similar to U4/U6.U5 tri-snRNP [50]. Of note, U1 and U2 exist as single particles that interact with RNA individually in the nucleus, and U11 and U12 form a U11/U12 di-snRNP, potentially through a protein-mediated interaction and interact with the pre-mRNA as a complex [59,60].

Protein composition of the minor spliceosome is yet another important aspect of the pathway that has received attention, leading to the identification of proteins that are unique to the minor spliceosome or shared with the major spliceosome. More than 300 protein have been identified in snRNP complexes, including snRNP-associated proteins (Sm proteins, Sm-like proteins, and snRNP-specific peptides) (Figure 3) and non-snRNP associated proteins (splicing factors) [14]. Interestingly, many of the snRNP-associated (i.e., Sm protein family) and non-snRNP-associated proteins (i.e., SR protein family) are shared between the minor and major spliceosomes [11,14,50,61]. For instance, U11/U12 di-snRNP has been shown to share all seven SF3b proteins with U2 snRNP, in addition to p14, hPrp43, YB-1 and Urp [14,21,62,63,64,65]. However, U11/U12 di-snRNP lacks all U1-specific proteins and instead contains unique proteins (65 K, 59 K, 48 K, 35 K, 31 K, 25 K, and 20 K), some of which are U11 specific [63]. This indicated that protein–protein and protein–RNA interactions at the 5′ ss are not conserved between the two spliceosomes [63]. Interestingly, these unique U11/U12 proteins are conserved in animals and plants [63,64,65]. On the other hand, the protein composition of U4atac/U6atac.U5 tri-snRNP is almost identical to that of U4/U6.U5 [50]. One notable shared protein, 15.5 K, binds the 5′ stem loop of U4 and U4atac snRNA and could play a nucleation factor role, allowing similar proteins to associate with U4/U6 and U4atac/U6atac [50,66,67,68]. However, unique minor tri-snRNP peptides remain to be identified [50]. The large number of proteins shared between the two spliceosome types suggest that protein–protein and protein–RNA interactions are similar or conserved to some extent.

The assembly of the minor spliceosome is very similar to that of the major spliceosome [13,58,60,68]. This is a stepwise process that involves the formation of multiple complexes (A, B, and C) through RNA–RNA, RNA–protein and protein–protein interactions, prior to the onset of catalysis [13,48,58,60,68] (Figure 1B). Early assembly involves the recognition of the 5′ ss and BPS cooperatively, through U11/5′ ss and U12/BPS interactions by the U11/U12 di-snRNP, which allows the formation of pre-spliceosomal complex A [13,58,60]. Subsequently, snRNA–snRNA base pairing allows the formation of U4atac/U6atac di-snRNP in which U4atac functions as an RNA chaperone to allow 5′ ss recognition by U6atac [48,58]. U4atac/U6atac and U5 are recruited to the spliceosome to form mature spliceosomal complex B [13,58]. The U4atac/U6atac.U5 tri-snRNP interaction disrupts the base pairing between U4atac/U6atac displacing U4atac before the first catalytic reaction [48,58]. U5 interacts with the exonic sequences at the 5′ ss and 3′ ss, and U6atac base pairs with the 5′ ss displacing U11 and interacting with U12 [48,58]. Consequently, U6atac and U12 form the activated catalytic core while U5 aligns the two exons for the second catalytic step, these form complex C that is catalytically active [13,48,58,68].

The identification of the snRNA and protein components in the minor spliceosome has helped shed light on the evolutionary relationship between the minor and major splicing systems. This relationship was described by the three models (previously discussed), which attempt to answer the question of whether these two systems originated from a common ancestor (homologous origin) [19]. The fact that most proteins in U4/U6.U5 are shared with U4atac/U6atac.U5, while many other proteins are shared between U2 and U11/U12 [50,61,65], supports a model of common ancestry [19]. However, the identification of novel unique U11/U12 proteins that have 5′ ss interactions, distinct from U1 [63], challenges the homologous origin model. This is further exacerbated by evidence of low percentage/lack of sequence conservation between minor and major spliceosomal snRNAs [26,58]. Therefore, a nonhomologous model such as a parasitic group II intron invasion model could explain the highly diverged snRNA and the high protein similarity in the minor and major spliceosomes [19,50,63]. However, this model does not explain the novel proteins identified in the minor spliceosome [63]. Instead, a homologous model with high divergence such as the fission fusion model could account for the current observations [19,63]. Speciation may have led to two separate lineages, each containing one splicing pathway in which snRNAs and spliceosomal proteins diverged separately [19,63]. Eventually, endosymbiosis allowed the fusion of the two lineages, forming an ancestral eukaryote with both splicing systems [19,63].

## 8. Splicing and Disease

The ability of splicing mutations to cause disease is well recognized through the myriad of described diseases that result from altered major or minor and/or alternative splicing [37,39,69]. Generally, these alterations in splicing patterns are a consequence of mutations to the cis-acting elements or the trans-acting factors. The cis-acting elements include the core splicing consensus sequences (5′ ss, 3′ ss, and BPS). In fact, 15% of disease-causing SNPs are located within the core splice sites. The other key set of cis-acting elements are splicing enhancers and silencers sequences also known as splicing regulatory elements (SREs). These could be within exons, exonic splicing enhancers (ESE), and exonic splicing silencers (ESS), or within introns, intronic splicing enhancers (ISE), and intronic splicing silencers (ISS) [70]. On the other hand, trans-acting factors include RNA-binding proteins, classified as spliceosomal and non-spliceosomal components that facilitate spliceosomal recruitment or suppression. Several disease-causing mutations affecting splicing through SREs or trans-acting elements have been identified [39,70,71,72], but many more are yet to be characterized, which undoubtedly would uncover a large set of disease causing mutations in the splicing realm. In addition, while most research attention has been focused on diseases affecting U2 intron splicing (review in [37]), a growing literature suggests that mi-INTs splicing alterations may be as important in disease [39,71,72,73,74,75].

## 9. Minor Intron Splicing and Diseases

The minor splicing pathway has been linked to three classes of disease, autoimmune, cancer/cancer predisposing conditions, and neurological (congenital and degenerative) [39,72,73,74,75] (Figure 1C and Figure 3 and Table 1). A testament to the essential role of mi-INTs, mutations that completely abolish their splicing are lethal [76,77]. For instance, complete loss of *Rnpc3*, which codes for 65 K, a protein unique to the minor spliceosome, causes death in mice embryos prior to blastocyst implantation [78]. Constitutive biallelic loss of *Rnu11* in mice is embryonically lethal [79]. Furthermore, Microcephalic Osteodysplastic Primordial Dwarfism type I, a disorder caused by homozygous or compound heterozygous *RNU4ATAC* mutations [80,81], albeit hypomorphic, causes early death in homozygous patients [82,83]. Another example is complete loss of SMN protein, which functions in snRNP maturation affecting miGs to a great extent [84], is lethal in mice, prior to implantation [76]. However, many of the mutations related to minor intron splicing cause partial loss, rather than complete loss of the minor spliceosome, or the function of miGs. This allows organismal survival and phenotypic manifestation of pathologies. Below is a brief description of the role of mi-INTs splicing in diseases.

A strong support for a role of splicing in autoimmune diseases comes from the observation that spliceosomal proteins (major and minor) have been shown to have important roles in systemic lupus erythematosus, polymyositis and diffuse scleroderma [101,102,103]. Interestingly, Ng et al. have shown a significantly higher rate of minor intron splicing in transcripts generating autoantigens as opposed to randomly selected genes (80% vs. 1%) [75].

The congenital disorders related to minor intron splicing include: spondyloepiphyseal dysplasia tarda (SEDT) [86], Microcephalic Osteodysplastic Primordial Dwarfism type I/III or Taybi-Linder syndrome (MOPDI/III or TALS) [89,104], Roifman syndrome (RFMN) [91], Lowry Wood syndrome (LWS) [90], early-onset cerebellar ataxia (EOCA) [87], isolated growth hormone deficiency (IGHD) [74] (reviewed in [39.72]), and Noonan syndrome [88]. On the other hand, the neurodegenerative diseases include spinal muscular atrophy (SMA) and amyotrophic lateral sclerosis (ALS) (reviewed in [39,72]). These diseases can be further classified by the mutation type that would lead to loss of function of the minor splicing system [39]. These involve defects in components of the minor snRNPs, mutation of U12-type 5′ ss, and defect in biogenesis and/or assembly of the minor spliceosome [39,72] (Figure 1C and Figure 2).

While deregulated alternative splicing in cancer is well established [10,105], most studies have focused on major intron splicing. However, the high similarity of the two pathways in their physiological states and the prevalence of alternative splicing in the minor splicing pathway [55] suggest that the two pathways are affected similarly in pathological states. Furthermore, the minor splicing system has been linked to myelodysplastic syndrome [71,88] and Peutz-Jegher’s syndrome [85], which increase the risk of acute myeloid leukemia [71] and gastrointestinal/extra gastrointestinal cancers [105], respectively. The cause of six congenital diseases and two neurodegenerative disorders have been attributed completely or in part to the minor splicing pathway.

One property of mi-INTs is their enrichment in information-processing genes. This creates a functional exclusiveness, crucial for cell survival and signaling (i.e., DNA transcription, replication and repair, RNA processing and translation, cytoskeletal organization, vesicular transport, voltage-gated ion channels, and Ras-Raf signaling) [22,23]. Therefore, based on the exclusivity of minor introns to certain gene families and their incomplete functional loss in diseases, it is highly likely that they are implicated in diseases such as cancer, where alterations in these gene families contribute to the hallmarks of cancer. Indeed, mi-INTs are enriched in cancer-relevant oncogenes such as *BRAF*, *ERK2*, *MAPK11/P38beta*, and *JNK1* [78,106]. Furthermore, p38MAPK, which has been implicated in mi-INTs splicing regulation [34], itself harbors an mi-INT. Other examples of cancer-relevant miGs include *PARP1*, which has many functions including DNA damage/repair [107] and the tumor suppressor *PTEN.* As shown in Figure 4, mi-INTs splicing deregulation in such genes could lead to consequences that may cause functional loss of the gene product or the production of an alternative isoform that could possibly act as dominant negative. In the absence of gene redundancy that could compensate partially for functional loss, the consequences could be genomic instability and chromosomal abnormalities [107] in addition to abnormal cell proliferation. We therefore propose that mutations affecting the minor splicing pathway would impact many pathways, and the culmination of effects from all miGs affected is thus expected to contribute to cancer pathogenesis.

## 10. Systemic vs. Tissue-Specific Pathologies

Theoretically, perturbation of mi-INTs splicing or ubiquitously expressed proteins involved in this splicing pathway should lead to multisystem pathologies, reflecting the ubiquitous nature of the etiology. However, in most diseases related to defects in mi-INTs splicing, only a single or a handful of tissues are affected. For instance, MOPD I/III, RFMN, and LWS which are caused by hypomorphic mutations in U4atac snRNA [81,90,91,104] lead to intrauterine growth restriction, poor growth, developmental abnormalities spanning the skeletal and nervous systems, and retinal abnormalities [83,90]. A clear mechanism behind these phenotypes has not been identified; however, it is suggested that they are certainly tied to abnormalities in mi-INTs splicing [79,91]. Another example is SMA, which manifests primarily in the nervous system and neuromuscular junctions, through motor neuron loss and muscle atrophy (Figure 5) [108,109]. This disease is caused by a mutation or homozygous deletion in *SMN1* with preservation of *SMN2* allowing the production of the SMN protein at low levels (Figure 5). SMN is essential for major and minor snRNPs biogenesis and assembly (Figure 6), a function that is essential in all cells of any tissue type. However, it has been shown that low SMN causes a biased decrease in minor snRNAs in SMA mice [94,95,96], causing splicing defects in some miGs [97,98,99,100]. In fact, selective enhancement of mi-INTs splicing in SMN-deficient mice, improved SMA disease phenotype, motor function, rescue of the loss of proprioceptive sensory synapses, and prolonged survival [84]. Thus, the ubiquitous nature of SMN and minor snRNPs implies that a deficiency would affect all tissues. Nevertheless, similar to MOPD I/III, RFMN, and LWS, SMA is also tissue specific. This dilemma has remained largely unresolved, but recent studies provide some possible explanations. Olthof et al. showed that miGs are differentially expressed in mice and human tissues, with tissues having the highest number of expressed miGs such as the cerebrum showing high expression levels [55]. Furthermore, the expressed miGs in all tissues are enriched in functions important for that specific tissue [55]. This suggests that tissues with highest number of expressed miGs, especially those with tissue-specific functions, are more sensitive to mi-INTs splicing deregulation. Indeed, 7 out of the 10 currently described mi-INTs-related diseases involve the nervous system; IGHD, EOCA, MOPD I/III, RFMN, LWS, ALS, and SMA. Furthermore, transcriptome analysis from MOPD I patients shows differential mi-INTs retention across tissues and miGs, again indicating that certain tissues and miGs are more susceptible/sensitive to mi-INTs retention. Consequently, the same mutation (i.e., *RNU4ATAC*) could cause a different mi-INTs retention levels in different tissues with certain miGs more affected than others. The biological determinants of mi-INTs retention susceptibility in tissues are not fully understood, but work by Doggett et al. shows that minor spliceosomal protein 65 K is localized to the stem/progenitor cell compartment, which is responsible for tissue maintenance in the small intestine [78]. Loss of this protein impaired minor intron splicing leading to apoptosis and decreased cell proliferation in the stem/progenitor cells population, with negligible effects on other small-intestine cell populations [78]. Similar findings were demonstrated by Baumgartner et al. in the context of microcephaly due to MOPD I, RFMN, and LWS. U11 loss in developing mice showed elevated mi-INTs retention contributing to preferential loss/apoptosis of a specific population of neural progenitor cells; the radial glial cell leading to microcephaly [79]. Most of mi-INTs retention led to NMD of the transcripts, the majority of which were enriched in functions such as nucleotide binding (e.g., *Parp1*) and protein transport and cell-cycle functions (e.g., *Pten*) [79]. Taken together, these findings demonstrate that within tissues, mi-INTs splicing and minor spliceosome components seem to be more important to certain cell populations than others. These cells are mostly those responsible for tissue development and maintenance, which could explain the high prevalence of developmental and cancer syndromes in mi-INTs spliceopathies.

Whether the cell population affected is a function of susceptible miGs/biological functions or vice versa is still unclear. On the one hand, the susceptibility of miGs to mi-INTs retention seems to be related to their splicing efficiency and expression levels in normal tissues, with those having low splicing efficiency and expression levels being more susceptible [82]. Another determinant of tissue specificity could be drawn from SMA pathogenesis. In this disease, minor snRNAs are decreased in brain, spinal cord, and heart tissue but remain unchanged in kidney and skeletal muscle tissues [95]. This differential expression of minor snRNAs could exist to some extent in normal tissues, which could be regulated in a tissue-specific manner and could potentially render tissues with the lowest snRNA expression at risk of higher rates of mi-INTs retention due to low minor snRNPs. This regulation could be at the level of transcription, biogenesis/assembly, or in the fully formed snRNP. Evidence for the latter has been documented in the fully formed U6atac, whose levels increase following stabilization by p38MAPK in cells [34]. Therefore, the idea of a possible regulatory mechanism that determines minor snRNP levels in healthy tissues and tissue susceptibility to mi-INTs retention is probable.

In the pathological state disruption of the SMN1 gene via a mutation or deletion, with preservation of SMN2 gene, lowers the levels of the SMN protein, decreasing the levels of the SMN complex and its ability to interact with Sm proteins. This leads to a preferential decrease in minor snRNPs. Ultimately, both the major and minor splicing pathways are affected with minor splicing defects being more frequent as a result of U12-type intron retention [95,99].

## 11. Minor Splicing and Disease Onset

The minor splicing system is important for tissue maintenance and renewal during development and adult life. In other words, this is a system that is indispensable for a normal physiology and life (as described above). The crucial role of this splicing system in development and adulthood bares the question of whether the same mutation affecting mi-INTs splicing could manifest differently depending on the age of onset. For instance, *Rnpc3* loss during the embryonic stage prevented implantation in mice, while in adult tissue, it affected the integrity of many tissues, especially the gastrointestinal tract [78]. On the other hand, the biallelic loss of *RNPC3* has been linked to IGHD in human [74]. Similarly, constitutive loss of U11 was embryonically lethal in mice but led to microcephaly when the loss occurred later in development [79]. Furthermore, U12-type intron splicing seems to be important for specific populations of cells namely stem/progenitor cells [78,79]. These findings allow us to speculate that age of onset may play an important role in determining the MiGs affected and consequently the phenotype observed. Under this premise, the loss of U11 later into adulthood may contribute to a neurodegenerative condition as opposed to microcephaly. Understanding the relationship between mutations that cause defects in mi-INTs splicing and age of onset may uncover more diseases that may be related to this splicing pathway. It would allow for a better appreciation of the differential regulatory roles that this pathway plays in developing and adult tissues. Furthermore, it would enhance our grasp of disease mechanisms related to this class of introns, which are still lacking.

## 12. Identification of Key Disease-Associated miGs

The current understanding of mi-INTs splicing-related diseases is limited. Importantly, whether the phenotypes observed are a consequence of a systemic splicing disruption or a subset of disrupted miGs remains unanswered. The tissue specificity of pathological conditions associated with mi-INTs splicing and its physiologic role in regulating gene expression and alternative splicing [34,35,39,55,56,78,79] suggest that only a subset of miGs is responsible for a given disease. Based on this, a number of in vitro and in vivo experiments could be designed to identify disease-causing miGs using CRISPR/Cas system and antisense oligonucleotides (ASO). An overall scheme of the approaches that are needed to move our understanding of mi-INTs splicing from the laboratory to the clinic is presented in Figure 7. This scheme relies initially on comprehensive methods such as RNA-seq to identify splicing changes, especially those in miGs, in a given disease. Splicing aberrations that are potentially relevant to the disease are then shortlisted for targeting. We propose two parallel approaches moving forward. (1) A CRISPR approach in which the miG or the specific intron is modified in the genome of cells extracted from patients or controls followed by testing the effects of this genome editing on disease progression. (2) ASOs on the other hand can modulate splicing at the RNA level without modifying the genome. A set of ASOs can be used to modulate several splicing events and studied in vitro to narrow it down to a handful that can be tested in an animal model. It is worth noting here that these ambitious approaches come with their own hurdles. These include but are not limited to issues with delivery to specific cells, in addition to off-target genome editing in the case of CRISPR or off-target splicing modulation in the case of ASOs. These hurdles necessitate in-depth follow-up studies before these approaches can be adapted as therapies in humans. One way to help mitigate these hurdles is creating animal models with aberrant splicing in miGs that mimic disease progression. These models can be used to further understand the relationship between altered mi-INTS splicing and disease, optimizing some of the limitations, and eventually drug testing and further preclinical trials testing. The identified genes along with the targeting methods used to identify them could have tremendous value in developing targeted therapies.

Targeted therapy has been successfully used in the treatment of many diseases. In cancer, targeted therapy has been used after identifying the gene or protein that are crucial to cancer progression and modulating them. However, with splicing being a targetable process, the list of targetable elements should include genes and mRNA transcripts that are underexpressed. The monogenic nature of mutations contributing to mi-INTs associated diseases makes these mutations good targets (i.e., SMA) [110]. The majority of mi-INTs defects result in mRNA transcripts retaining mi-INTs, which, if they escape NMD, may cause the production of a novel protein isoform or nonfunctional protein. The occurrence of such an event in the tumor suppressor *PTEN*, for instance, could be detrimental. However, the use of ASOs or splice switching oligonucleotides (SSOs) [111] have the potential to promote adequate splicing in the setting of mi-INTs retention. A successful application of splice site targeting was done using CRISPR-guided cytidine deaminase (TAM) to restore the open reading frame of a mutant *DMD* gene [112]. The ability of the TAM system to modify splice sites could potentially have a role in the treatment of mi-INTs splice site mutations such as those in *STK11* and *TRAPPC2*. These methods provide a broad spectrum of approaches by which mi-INTs splicing could be targeted through correcting the main causative mutation or the splicing of affected MiGs.

## Figures and Tables

**Figure 1 ijms-22-06062-f001:**
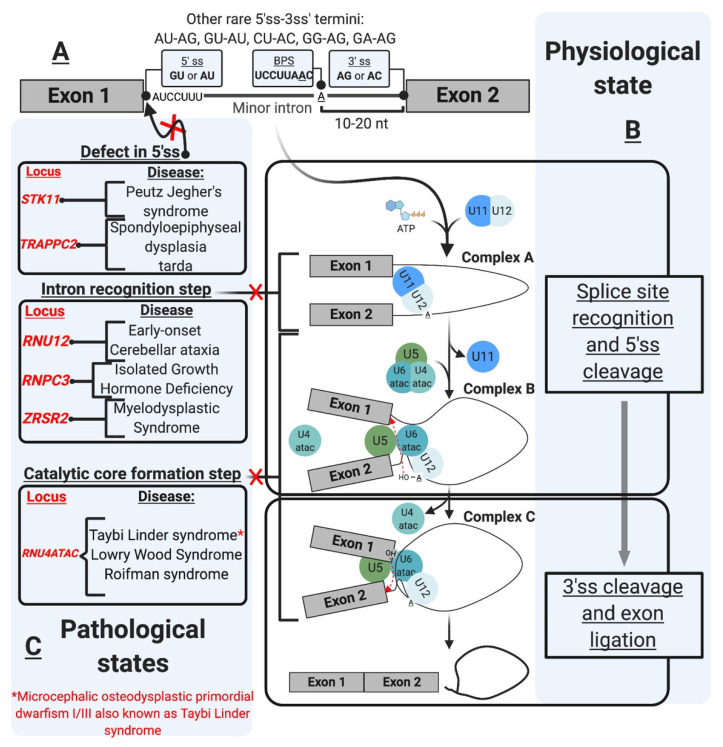
The minor intron splicing pathway. (**A**). U12-type intron sequence characteristics [1,2,3,4,5,6]. These sequences were initially described in the literature based on DNA nomenclature. These were changed in the figure as the transcript depicted is an RNA transcript. The commonly observed 3′ and 5′ sequences of minor introns are shown in the boxes. The conserved sequences are shown at the 5′ ss and branchpoint (**B**). Spliceosomal assembly and the splicing reaction [7,8,9,10,11,12] occur as a two-step reaction. The interaction of the preformed U11/U12 di-snRNP leads to the formation of Complex A. U4atac/U6atac.U5 tri-snRNP allows the formation of Complex B, which carries the first transesterification reaction after the release of U4atac and U11. This reaction leads to the formation of Complex C, which carries the second transesterification reaction producing ligates exons and a minor intron lariat. (**C**). U12-type intron-associated diseases classification [13,14,15,16,17,18,19,20,21]. The diseases are depicted in association with the affected step. Diseases not shown in the figure include amyotrophic lateral sclerosis associated with mutations in fused in sarcoma (FUS) RNA-binding protein [22], and Noonan syndrome due to U12-type intron retention in LZTR1, a regulator of RAS-related GTPases [23].

**Figure 2 ijms-22-06062-f002:**
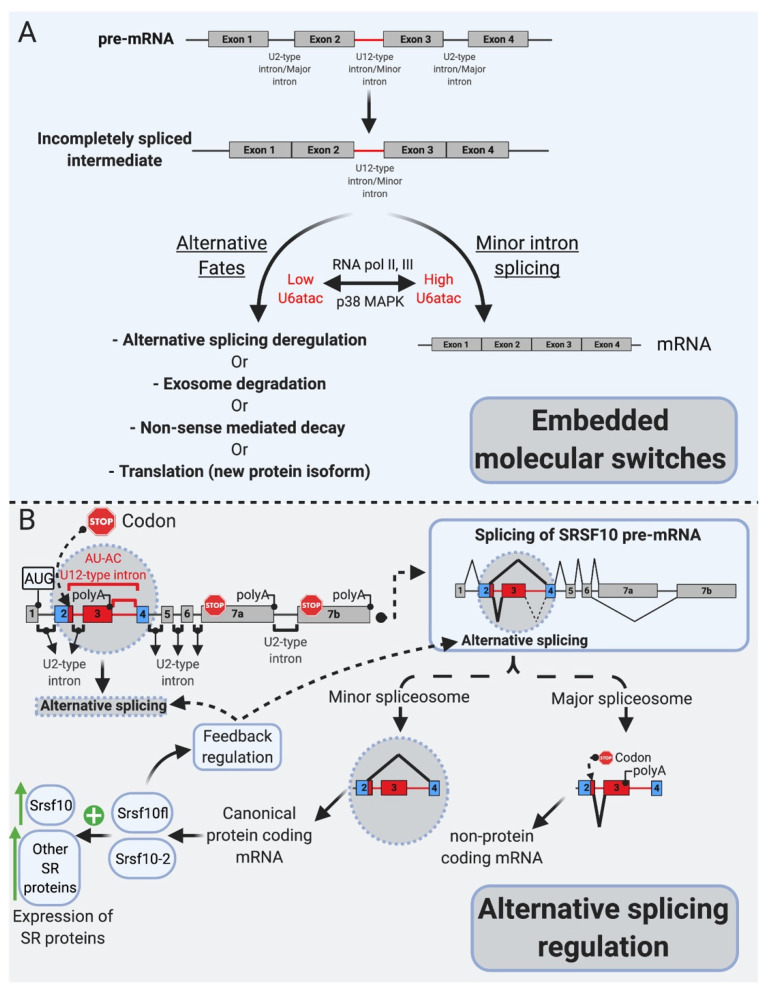
mi-INTs physiologic roles. (**A**). mi-INTs function as embedded molecular switches of gene expression [25]. Instability of U6atac result in low U6atac levels, which leads to pre-mRNA transcripts with a retained mi-INT whose fates are depicted in Figure 3. Decreased transcription lowers U6atac levels and the expression of minor-intron-containing genes, while activated p38MAPK stabilizes U6atac, increasing its levels and allowing the proper splicing of mi-INTs, producing a full-length mRNA. (**B**). mi-INTs regulate alternative splicing through Srsf10 [24]. Low minor spliceosome activity causes an increase in the expression of non-protein-coding Srsf10 transcript through exon 3 inclusion as a consequence of major intron splicing. High minor spliceosome activity allows exclusion of exon 3, through splicing to exon 4, allowing formation of a protein-coding transcript. The functional Srsf10 autoregulates its expression and that of other SR proteins which regulate alternative splicing of thousands of downstream targets.

**Figure 3 ijms-22-06062-f003:**
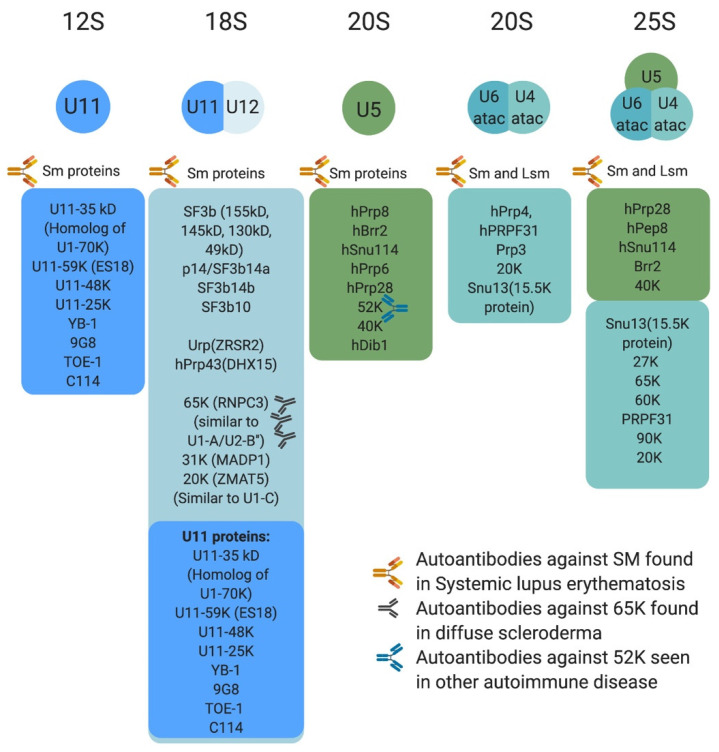
Protein composition of minor snRNPs and autoimmune conditions associated with them. Sm proteins include B/B’, D3, D2, D1, E, F, and G, whereas LSm proteins are LSm 2–8. The U4atac/U6atac.U5 tri-snRNP has two sets of Sm protein and one set of LSm proteins. Antibodies against some protein components of the minor spliceosome are found in systemic lupus erythematosus, diffuse scleroderma, and other autoimmune conditions.

**Figure 4 ijms-22-06062-f004:**
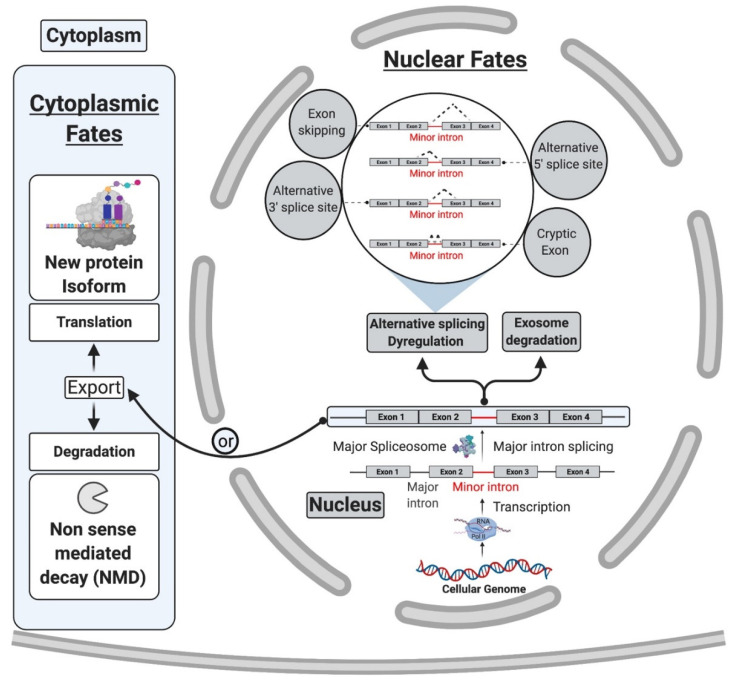
Cytoplasmic and nuclear fates of pre-mRNA transcripts with retained mi-INTs [20,26,27]. mi-INTs retention may lead to trapping of the transcript within the nucleus. The trapped transcript may undergo exosomal degradation, activation of cryptic U2-type intron cryptic sites or exon skipping. Transcripts that escape degradation in the nucleus are exported to the cytoplasm they may be degraded by NMD or translated to a truncated or novel protein isoform.

**Figure 5 ijms-22-06062-f005:**
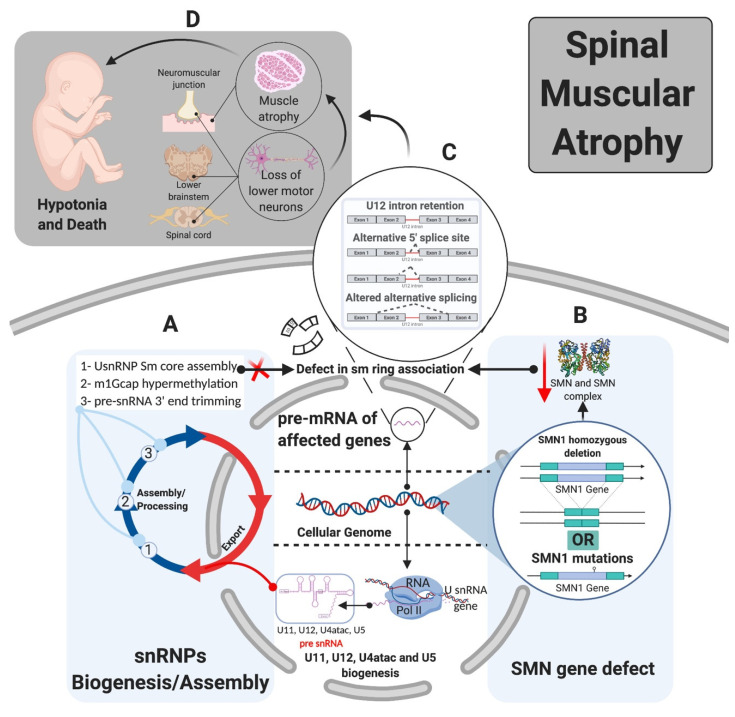
Tissue specificity of mi-INTs-related diseases (SMA). (**A**). snRNP biogenesis and assembly requires SMN along with other proteins (SMN complex) for Sm core assembly, m1Gcap hypermethylation, and pre-snRNA 3’end trimming. (**B**). Loss or mutation in SMN1 with retention of SMN2 product allows the production of SMN at lower levels leading to lower SMN complex level and lower ability to interact with Sm proteins, preventing the formation of the Sm ring and mature snRNPs (major and minor) which preferentially affects minor snRNPs [28,29,30] leading to splicing defects. (**C**). Splicing defect in U2 and U12-type intron splicing with bias toward U12-type intron retention [29,31] leads to aberrant mRNA transcripts that may be degraded or produce aberrant proteins. (**D**). Phenotype of SMA depends on the severity of the illness, which is caused by motor neuron degeneration in the brain stem, anterior horn of the spinal cord, and muscle atrophy. This leads to hypotonia and eventual death [32,33].

**Figure 6 ijms-22-06062-f006:**
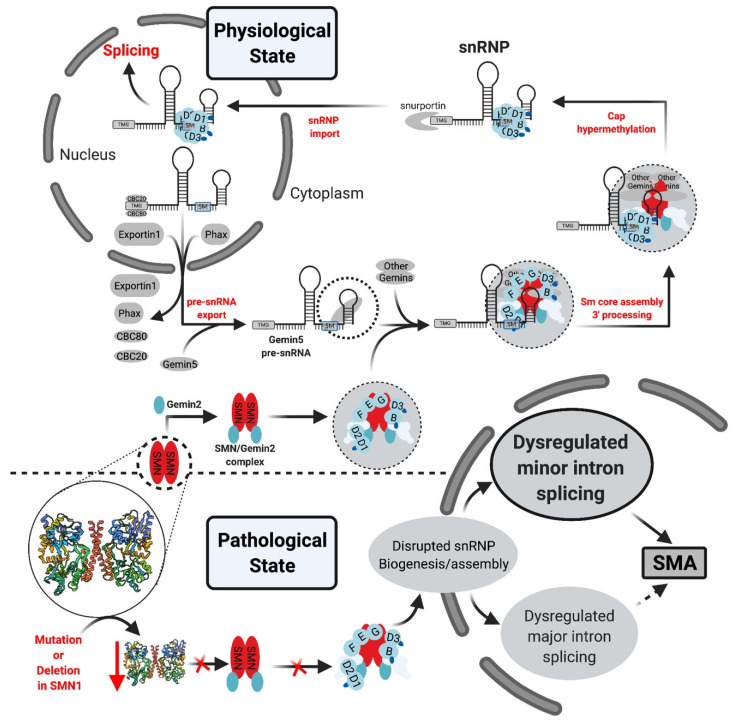
U snRNP biogenesis pathway. In the physiological state, once the U snRNA is transcribed and capped it associates with multiple factors which form the export complex and allow U snRNA export into the cytoplasm. Subsequently, in a process mediated by SMN complex and Gemin proteins, the Sm proteins interact with the U snRNA at the Sm site. The SMN complex dissociates, followed by 3′ processing/trimming and cap hypermethylation. The final product is imported into the nucleus. Further modifications occur on the U snRNA, and snRNP (U1, U2, U4, U5 or U11/U12, U4atac)-specific proteins associate with U snRNP Sm core [34].

**Figure 7 ijms-22-06062-f007:**
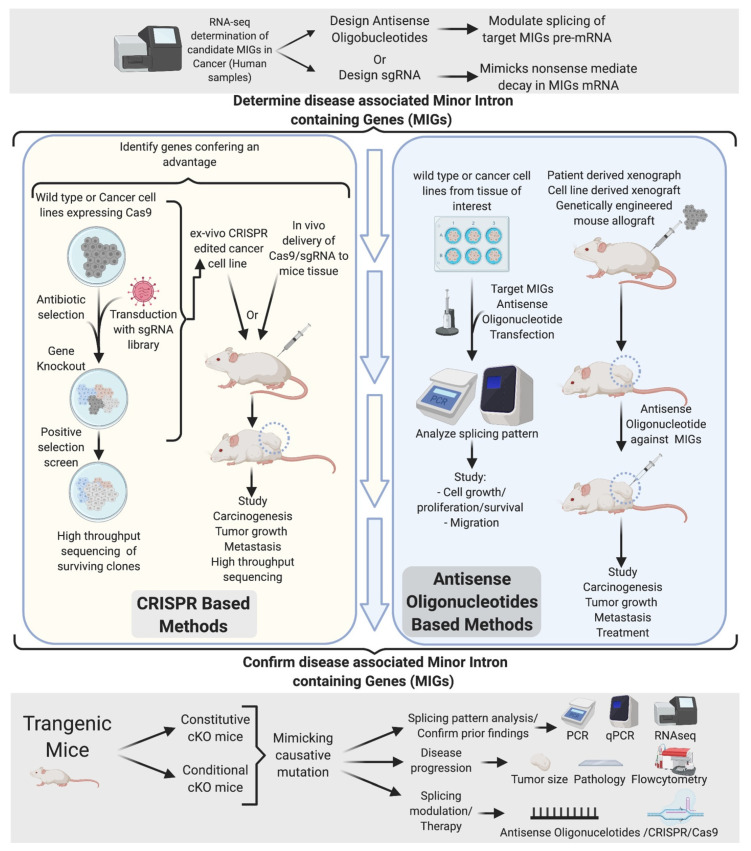
Experimental determination of disease-causing/-associated MIGs. Initial determination of disease-related MIGs through RNA-seq generates a list of potential candidates. Targeting of these candidates through CRISPR and ASOs in vivo and vitro narrows down the gene targets to a few. The ultimate confirmation comes from a genetically engineered mouse, mimicking the disease mutation found in human. This allows confirmation of mi-INTs splicing defects observed and determination of their contribution to the disease.

**Table 1 ijms-22-06062-t001:** Human diseases associated with defects in minor intron splicing.

Splicing Defect	Gene Affected	Disease	References
Aberrant 5’ splice site	*STK11*	Peutz Jegher’s syndrome (PJS)	[85]
	*TRAPPC2*	Spondyloepiphyseal dysplasia tarda (SEDT)	[86]
Intron recognition/mutant minor snRNP component	*RNU12*	Early-onset cerebellar ataxia (EOCA)	[87]
	*RNPC3*	Isolated growth hormone deficiency (IGHD)	[74]
	*ZRSR2*	Myelodysplastic syndrome (MDS) and acute myeloid leukemia (AML)	[88]
	*RNU4ATAC*	Taybi-Linder syndrome (TLS)	[89]
	*RNU4ATAC*	Lowry Wood syndrome (LWS)	[90]
	*RNU4ATAC*	Roifman syndrome	[91]
Branch point mutation	*LZTR1*	Noonan syndrome (NS)	[88]
	*LZTR1*	Schwannomatosis	[92]
Biogenesis/Assembly	*FUS*	Amyotrophic lateral sclerosis (ALS)	[93]
	*SMN1*	Spinal muscular atrophy (SMA)	[94,95,96,97,98,99,100]

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
