# Peer review of "Minor Intron Splicing from Basic Science to Disease"

_ijms, 2021, doi:10.3390/ijms22116062_

Round 1

Reviewer 1 Report

ijms-1226218

Minor Intron Splicing from Basic Science to Disease

Ettaib El Marabti, Joel Malek and Ihab Younis

In this review article, Marabti et al describe the evolutionarily conserved minor introns with basic mechanism and related diseases. They nicely introduce minor introns regulatory roles in alternative splicing and minor spliceosome-associated diseases with tissue specific pathologies. The authors clearly describe their importance as a key regulatory pathway, could lead to tissue specific pathologies due to specific alterations in the expression of some minor intron containing genes.

This review article is well prepared and beneficial to both basic scientists and physicians who are interested in minor-intron deficient diseases.

I have some minor comments and suggestions described below.

1) It would be nicer if the authors could have a table to show compilation of some minor intron sequences for 5’ss, bp and 3’ss from some species.

2) It would be better to include ‘twintron’ case for the review. Prospero is the representative of this case.

3) A table with diseases and their responsible minor intron containing genes would be so helpful to the readers.

Author Response

We are grateful for the reviewer’s positive summary of the manuscript. We could not have agreed more with the reviewer’s description of the topic and its impact.

1) It would be nicer if the authors could have a table to show compilation of some minor intron sequences for 5’ss, bp and 3’ss from some species.

We thank the reviewer for this comment, and we agree that such a table would be quite helpful but given that the information is already available on an online database (https://genome.crg.es/cgi-bin/u12db/u12db.cgi) we opted to reference this database rather than recreate it in the review. This is now added on page 7.

2) It would be better to include ‘twintron’ case for the review. Prospero is the representative of this case.

We agree with the reviewer. We elaborate more on the prospero twintron on page 8.

3) A table with diseases and their responsible minor intron containing genes would be so helpful to the readers.

We thank the reviewer for this suggestion. While the information is spread in multiple figures, we agree that a table would be quite helpful. We have added this table to the revised manuscript. We reference the Table on page 16.

Reviewer 2 Report

It is a well written review on minor intron splicing and related diseases. My major concern is the last section "Identification of key diseases associated miGs" too brief. Need more elaborations on (1) the successful cases end up with the strategy in Fig. 7 to identify disease-causing miGs and (2) what are the potential hurdles for detection and treatment.

Author Response

(1) the successful cases end up with the strategy in Fig. 7 to identify disease-causing miGs

We thank the reviewer for taking the time to review our manuscript, and we appreciate the insightful feedback. We have kept this section purposely short because the approaches in Figure 7 are merely suggestions on how to move the field forward. Having said so, we elaborated this section further on pages 20 and 21. As for successful examples of this strategy, we relied on other fields such as cancer therapeutics, where the CRISPR approach is now in clinical trials. We also have in the text examples of successful usage of ASO in SMA and CRISPR in DMD. We believe that this could be adapted for minor introns-related diseases.

(2) what are the potential hurdles for detection and treatment.

As for the hurdles, we do anticipate many as is the case for any ambitious approach like the one we suggested. We believe though that if we start discussing hurdles in too many details, the manuscript will become too technical and would deviate from the original spirit of putting together a comprehensive review of minor intron splicing. However, we did add a part that addresses potential hurdles such as delivery to specific cells and off-targets editing in the case of CRISPR or off-target splicing modulation in the case of ASOs. We also suggested in-depth follow-up studies before they can be adapted for human therapy. These are now on page 21.